# ESTIMATING EXAMPLE DIFFICULTY USING VARIANCE OF GRADIENTS

## ABSTRACT

In machine learning, a question of great interest is understanding what examples are challenging for a model to classify. Identifying atypical examples helps inform safe deployment of models, isolates examples that require further human inspection, and provides interpretability into model behavior. In this work, we propose Variance of Gradients (VoG) as a valuable and efficient proxy metric for detecting outliers in the data distribution. We provide quantitative and qualitative support that VoG is a meaningful way to rank data by difficulty and to surface a tractable subset of the most challenging examples for human-in-the-loop auditing. Data points with high VoG scores are far more difficult for the model to learn and over-index on corrupted or memorized examples.

## 1 INTRODUCTION

Reasoning about model behavior is often easier when presented with a subset of data points that are relatively more difficult for a trained model to learn. This not only aids interpretability through case based reasoning (Kim et al., 2016; Caruana, 2000; Hooker et al., 2019), but can also be used as a mechanism to surface a tractable subset of atypical examples for further human auditing (Leibig et al., 2017; Zhang, 1992; Hooker et al., 2019), for active learning to inform model improvements, or to choose not to classify certain examples when the model is uncertain (Bartlett & Wegkamp, 2008; Cortes et al., 2016).

One of the biggest bottlenecks for human auditing is the large scale size of modern datasets and the cost of annotating each feature (Veale & Binns, 2017). Methods which automatically surface a subset of relatively more challenging examples for human inspection help prioritize limited human annotation and auditing time. Despite the urgency of this use-case, ranking examples by difficulty has had limited treatment in the context of deep neural networks due to the computational cost of ranking a high dimensional feature space. Recent work in this direction has either been limited to small scale datasets or features a computational cost which is infeasible for most practitioners (Hooker et al., 2019; Carlini et al., 2019; Koh & Liang, 2017).

In this work, we start with a simple hypothesis – examples that a model has difficulty learning will exhibit higher variance in gradient updates over the course of training. On the other hand, we expect the backpropagated gradients of the samples that are *relatively easier* to learn will have lower variance because performance on that example does not consistently dominate the loss over the course of training. The gradient updates for the relatively easier examples are expected to stabilize early in training and converge to a narrow range of values. We term this class normalized ranking mechanism *Variance of Gradients* VoG, and demonstrate across a variety of large scale datasets that it efficiently ranks the difficulty of both training and test examples. VoG can be computed using either the predicted or true label, making it a valuable unsupervised auditing tool at test time when the true label is unknown.

**Validating the behavior of VoG on artificial data.** To begin, we illustrate the principle and effectiveness of VoG in a contrived toy example setting. The data was generated using two separate isotropic Gaussian clusters with a total of 500 data points. In such a simple low dimensional problem, the most challenging examples for the model to classify are closer to the decision boundary. In Fig. 1a we visualize the trained decision boundary of a multiple layer perceptron (MLP) with a single hidden layer trained for 15 epochs. VoG is computed at relative intervals for each training data point.

In Fig. 1b, we plot the final VoG score (Sec. 2) against the distance to the trained boundary. As expected, VoG allocates the highest scores to the most challenging examples that are closest to the decision boundary which exhibit the greatest variance in gradient updates over the course of the training process.

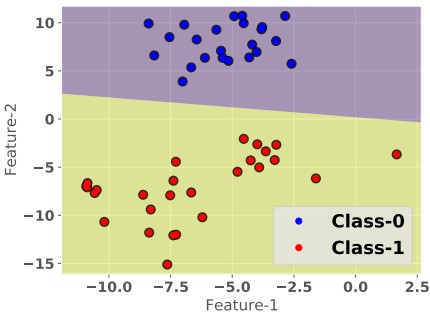

(a) Toy dataset trained decision boundary

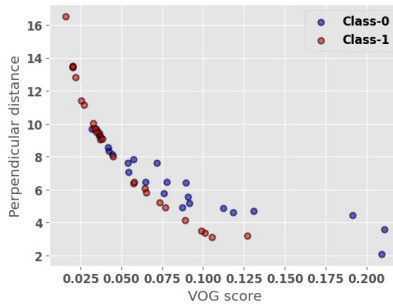

(b) Distance vs. VOG score

Figure 1: We compute the variance of gradients (VoG) for each training data point in this two dimensional toy problem. On the right, we show that VoG accords higher scores to the most challenging examples closest to the decision boundary (as measured by the perpendicular distance).

**Contributions** We scale this toy experiment and demonstrate consistent results across two different architectures and three datasets – Cifar-10, Cifar-100 Krizhevsky et al. (2009) and ImageNet (Russakovsky et al., 2015). Our contributions can be enumerated as follows:

1. **We propose Variance of Gradienst (VoG)** – a class-normalized variance gradient score for determining the relative ease of learning data samples within a given class (Sec. 2).
2. We show that VoG is an effective auditing tool for ranking high dimensional datasets by difficulty. VoG assigns higher scores to test-set examples that are more challenging for the model to classify. Restricting evaluation to the test-set examples with the lowest VoG greatly improves generalization performance. (Sec. 3).
3. **VoG identifies clusters of images with clearly distinct semantic properties.** As seen in Fig. 4), Low scores feature images with far less cluttered backgrounds and more prototypical vantage points of the object. In contrast, images with high scores over-index on images with cluttered backgrounds and atypical vantage points of the object of interest (zoomed in on part of the object, side profile of the object, shot from above).
4. **VoG effectively surfaces OOD and memorized examples** We empirically show that VoG allocates higher scores to examples that require *memorization* (Sec. 5) and out-of-distribution examples from curated benchmarks like ImageNet-O (Hendrycks et al., 2019). We use VoG to explore how learning differs at different stages of training and show that VoG rankings are sensitive to the stage of training and provide insight into the learning process in deep neural networks.

**Implications of this work** It is becoming increasingly important for deep neural networks (DNNs) to make decisions that are interpretable to both researchers and end-users. In sensitive domains such as health care diagnostics (Xie et al., 2019; Gruetzemacher et al., 2018; Badgeley et al., 2019; Oakden-Rayner et al., 2019), self-driving cars (NHTSA, 2017) and hiring (Dastin, 2018; Harwell, 2019) providing tools for domain experts to audit models is of upmost importance. Our work offers an efficient method to rank the global difficulty of examples and automatically surface a possible subset to aid human interpretability. VoG can be computed using checkpoints stored over the course of training and is model agnostic. Critically, VoG can be computed using the predicted label, which makes it an unsupervised auditing tool at test time.

## 2 METHODOLOGY

We consider a supervised classification problem where a DNN is trained to approximate the function $F$ that maps an input variable $X$ to an output variable $Y$, formally $F : X \mapsto Y$. $y \in Y$ is a discrete label vector associated with each input $x$. Each label $y$ corresponds to one of $C$ categories or classes.

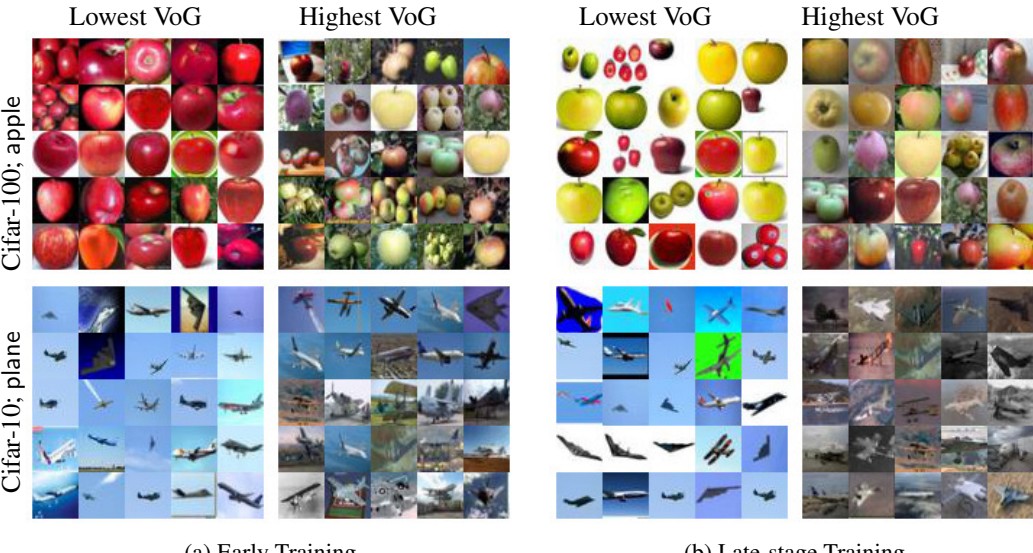

|            Lowest VoG            Highest VoG            Lowest VoG            Highest VoG            |

(a) Early Training           (b) Late-stage Training

Figure 2: The 5×5 grid shows the top-25 Cifar-10 and Cifar-100 training-set images with the lowest and highest VoG scores in the *Early* (a) and *Late* (b) training stage respectively of two randomly chosen classes. Lower VoG images evidence uncluttered backgrounds (for both apple and plane) in the *Late* training stage. VoG also appears to capture a color bias present during the *Early* training stage for both apple (red). The VoG images in *Late* training stage present unusual vantage points, with images where the frame is zoomed in on the object of interest.

A given input image $X$ can be decomposed into a set of pixels $x_i$, where $i = \{1, \ldots, N\}$ and $N$ is the total number of pixels in the image. For a given image, we compute the gradient of the activation $A_p^l$ with respect to each pixel $x_i$. Here, $l$ designates the pre-softmax layer of the network and $p$ is the index of either the true or predicted class probability. We consider $S$ as a matrix that represents the gradient of $A_p^l$ with respect to individual pixels $x_i$, *i.e.*, for an image of (say) $3 \times 32 \times 32$ size, $S$ will be a $3 \times 32 \times 32$ gradient matrix.

$$S = \frac{\partial A_p^l}{\partial x_i} \tag{1}$$

This formulation may feel familiar as it is often computed based upon the weights of a trained model and visualized as a image heatmap for interpretability purposes (Baehrens et al., 2010; Simonyan et al., 2013). Here, we instead intend to compute the average variance of the input gradients for the same image across training to arrive at a scalar score that is a proxy measure of how challenging the example is to learn. Without loss of generality, we take the sum across the color channels to arrive at a gradient matrix $S$ where $S \in \mathbb{R}^{32 \times 32}$. For a given set of $K$ checkpoints, we generate the above gradient matrix $S$ for all individual checkpoints, *i.e.*, $\{S_1, \ldots, S_K\}$. We then calculate the mean gradient $\mu$ by taking the average of the $K$ gradient matrices. Note, $\mu$ is the mean across different checkpoints and is of the same size as the gradient matrix $S$. We then calculate the variance of gradients across each pixel using the equations:

$$\mu = \frac{1}{K} \sum_{t=1}^{K} S_t \tag{2}$$

$$VoG_{pixel} = \sqrt{\frac{1}{K} \sum_{t=1}^{K} (S_t - \mu)^2} \tag{3}$$

Here, $VoG_{pixel}$ is a matrix representing the variance of gradients of each pixel in the image. We average the pixel-wise variance of gradients to compute a scalar VoG score for the given input image:

$$VoG = \frac{1}{N} sum(VoG_{pixel}) \tag{4}$$

where, $N$ is the total number of pixels in a given image. Hence, for every data sample $X$ we compute a scalar value indicating the variance of gradients score. To calculate the class-normalized VoG

score, we calculate the mean and deviation of all the VoG scores belonging to each class $c$, where $c \in \{1, \ldots C\}$, from the dataset. In order to account for inherent differences in variance between classes, we normalize the absolute ranking of the VoG score by class-level mean and standard deviation. This amounts to asking: *What is the variance of gradients for this image relative to all other exemplars for this class category?*

## 2.1 EXPERIMENTAL SETUP

**Datasets:** We evaluate our methodology on Cifar-10 and Cifar-100 Krizhevsky et al. (2009) and ImageNet (Russakovsky et al., 2015).

**Cifar Training:** We use a ResNet-18 network (He et al., 2016) for both Cifar-10 and Cifar-100. For each dataset, we train for 350 epochs using stochastic gradient descent (SGD) and compute the input gradients for each sample every 10 epochs. We implemented standard data augmentation by applying cropping and horizontal flips of input images. We use a base learning rate schedule of $0.1$ and adaptively change to $0.01$ at $150^{th}$ and $0.001$ at $250^{th}$ training epochs. The top-1 test-set accuracy for Cifar-10 and Cifar-100 were $89.57\%$ and $66.86\%$ respectively.

**ImageNet Training:** We use a ResNet-50 (He et al., 2015) trained on ImageNet. The network was trained with batch normalization (Ioffe & Szegedy, 2015), weight decay, decreasing learning rate schedules, and augmented training data. We train for $32,000$ steps (approximately $90$ epochs) on ImageNet with a batch size of $1024$ images. We store $32$ checkpoints over the course of training, but in practice observe that VoG ranking is very stable computed with as few as 3 checkpoints. Our model achieves a top-1 accuracy of $76.68\%$ and top-5 accuracy of $93.29\%$.

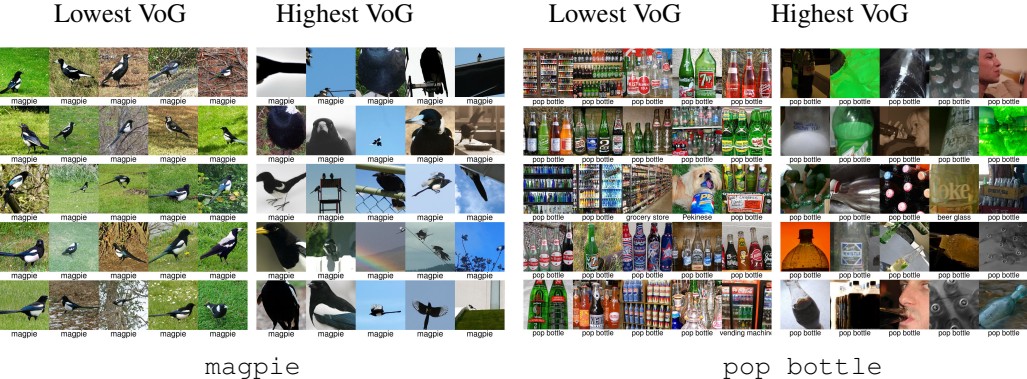

Figure 3: Each $5\times5$ grid shows the top-25 ImageNet training-set images with the lowest and highest VoG scores for the class `magpie` and `pop bottle` with their predicted labels below the image. Training set images with higher VoG scores (b) tend to feature zoomed-in images with atypical color schemes and vantage points.

## 3 EVALUATING VOG

For all datasets considered, we compute VoG for both training and eval sets. In this section, we evaluate the utility of VoG as an auditing tool. We evaluate the stability of the VoG ranking, measure how discriminative it is at separating easy examples from difficult, and comment on the qualitative properties of images at either end of the VoG spectrum.

**Qualitative inspection of ranking** A qualitative inspection of examples with high and low VoG scores shows that there are distinct properties to the images at either end of the ranking. We visualize 25 images ranked lowest and highest according to VoG for both the entire dataset (visualized for ImageNet in Fig. 5) and for specific classes (visualized for ImageNet in Fig. 3 and for Cifar-10 and Cifar-100 in Fig. 2). We observe that ranking by VoG produces clusters with clearly distinguished semantic features. Images with low VoG score tend to have uncluttered and often white backgrounds with the object of interest centered clearly in the frame. Images with the *highest* VoG scores have cluttered backgrounds and the object of interest is not easily distinguishable from the background. We also note that images with high VoG score tend to feature atypical vantage points of the objects

such as highly zoomed frames, side profiles of the object or shots taken from above. Other, the object of interest is partially occluded or there are image corruptions present such as heavy blur.

**Test-set error and VoG** A valuable property of an auditing tool is to be able to effectively discriminate between easy and challenging examples. Here, we measure whether VoG is able to do so. In Fig. 4, we plot the test-set error of examples bucketed by VoG decile. For this and the remainder of the experiments, we compute VoG using checkpoints stored from the first and last 3 epochs. Thus, at each point of the x-axis, we are computing the test-set error on the 10% of data whose VoG score falls between each decile. Note that we plot error, so lower is better. We show that examples at the lowest percentiles of VoG have far lower error rates. Mis-classification increases with an increase in VoG scores. Our results are consistent across all datasets, yet more pronounced for the more complex datasets Cifar-100 and ImageNet. We ascribe this to differences in underlying model complexity. In Fig. 11 we observe that test-set accuracy on the lowest VoG scored images improves beyond baseline test-set performance. Our models show improved generalization when restricted to low VoG images.

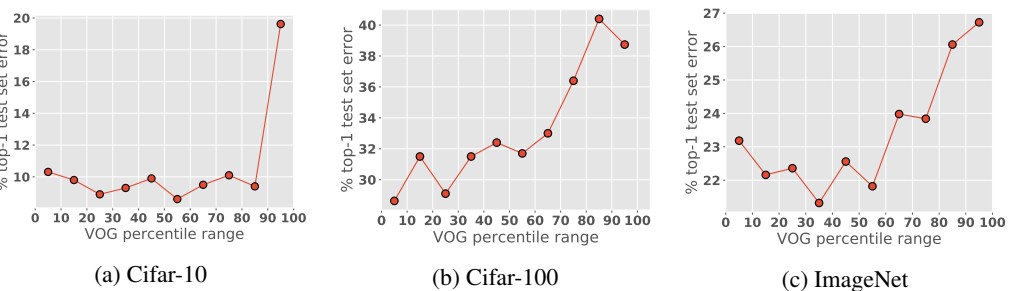

(a) Cifar-10          (b) Cifar-100          (c) ImageNet

Figure 4: The mean top-1 test set error (y-axis) for the exemplars thresholded by VoG score percentile (x-axis). Across Cifar-10, Cifar-100 and ImageNet, we observe that misclassification increases with an increase in VoG scores. Across all datasets we observe that the group of samples in the top-10 percentile VoG scores have the highest error rate, *i.e.*, contains most number of misclassified samples. For all datasets, model generalization improves on the bottom 10th percentile relative to the entire dataset.

**VoG as an unsupervised auditing tool** Many auditing tools to evaluate and understand possible model bias require the presence of labels for protected attributes and underlying variables. However, this is highly infeasible in real-world settings (Veale & Binns, 2017). For image and language datasets, the high dimensionality of the problem makes it hard to identify a priori what are underlying variables to be aware of. Even acquiring the labels for a limited number of attributes protected by law (gender, race) is expensive and/or may be perceived as intrusive leading to noisy or incomplete labels. One key advantage of VoG is that we show it continues to produce a useful ranking even when the gradients are computed with respect to the predicted label. In Fig. 5, we include the top and bottom 25 VoG ImageNet test images using predictions. We also computed the mean test-error for the predicted VoG distribution, and find that it also effectively discriminates between top-10 and bottom-10 examples with 73.6% and 77.5% accuracy respectively (Fig. 12).

**Stability of VoG ranking** To build trust with the end-user, a key desirable property of any auditing tool is consistency in performance. We would expect a consistent method to produce a ranking with a closely bounded distribution of scores across independently trained runs of the same model architecture and dataset. To measure the consistency of the VoG ranking, we train 5 Cifar-10 networks from random initialization using the training methodology explained in Sec. 2.1. Empirically, Fig. 6a shows that VoG rankings evidence a consistent distribution of test-error at each percentile given the same model and dataset.

# 4  LEVERAGING VOG TO UNDERSTAND EARLY AND LATE TRAINING DYNAMICS

Recent work has shown that there are distinct stages to training in deep neural networks (Achille et al., 2017; Jiang et al., 2020; Mangalam & Prabhu, 2019; Faghri et al., 2020). In our second set of experiments, we explore whether rankings according to VoG are sensitive to the stage of the

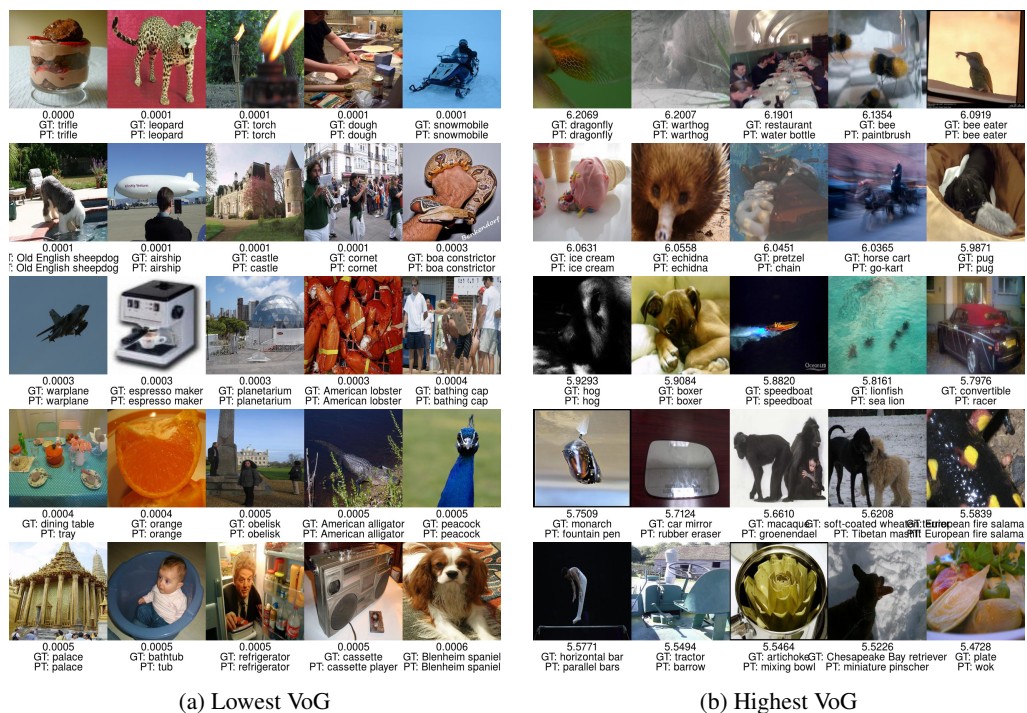

(a) Lowest VoG        (b) Highest VoG

Figure 5: Each 5×5 grid shows the top-25 ImageNet test-set images with the lowest and highest VoG scores for the top-1 predicted class. Test set images with higher VoG scores tend to feature zoomed-in images and are misclassified more as compared to the lower VoG images which tend to feature more prototypical vantage points of objects.

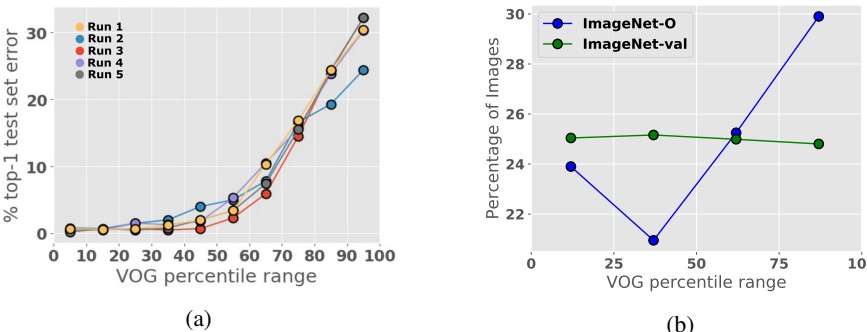

(a)            (b)

Figure 6: **Left:** Consistency in ranking is an important attribute of any auditing tool. Here, we plot the VoG top-1 test set error for 5 ResNet-18 networks independently trained on Cifar-10 from random initialization. The plots show that VoG produces a stable ranking with a similar distribution of error in each percentile across all images. **Right:** We measure the distribution of ImageNet-O images across percentiles. We find that higher percentiles of VoG over-index on these out of distribution images.

training process. Hence, we compute VoG separately for two different stages of the training process, which we term (1) the *Early* stage (first three epochs), and (2) the *Late* stage (last three epochs). Test-set accuracy at the *Early* stage is $44.65\%$, $14.16\%$ and $51.87\%$ for Cifar-10, Cifar-100 and ImageNet respectively. In the *Late* stage it is $89.57\%$, $66.86\%$ and $76.69\%$ for Cifar-10, Cifar-100 and ImageNet respectively.

We find that there is a noticeable visual difference between the image ranking computed for *Early* and *Late* stages of training. As seen in Fig. 2, for some classes such as `apple` it appears that VoG scores also capture network color bias present during the *Early* training stages. For these classes, the lowest VoG scores over-index on red colored apples.

For ImageNet, we compute the distribution of error in early and late stage and find a remarkable flipping point (Fig. 7). During the early training stage, samples having higher VoG score tend to have a lower error rate as the gradient updates center on easy examples. This phenomenon is reversed during the late stage of the training where most easy example have been learnt, and updates to the harder examples dominate the the computation of variance. Hence, samples having high VoG results in higher error rate.

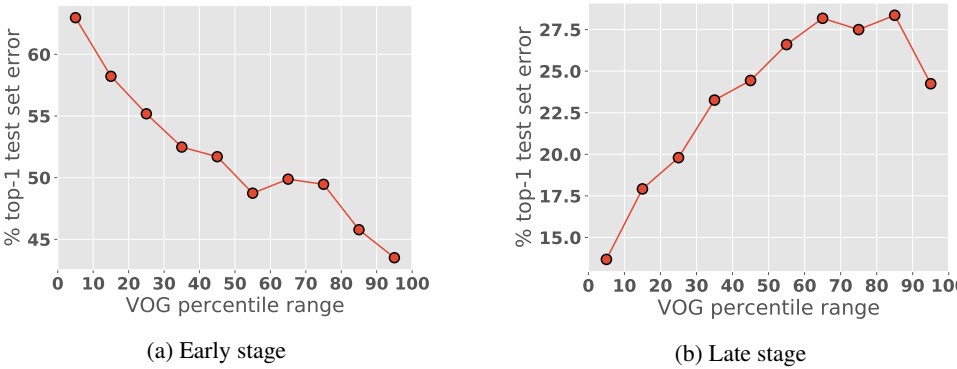

(a) Early stage                    (b) Late stage

Figure 7: The mean top-1 test set error (y-axis) for the exemplars thresholded by VoG score percentile (x-axis) in ImageNet validation set. The Early (a) and Late (b) stage VoG analysis shows inverse behavior where the role of VoG flips as the training progresses.

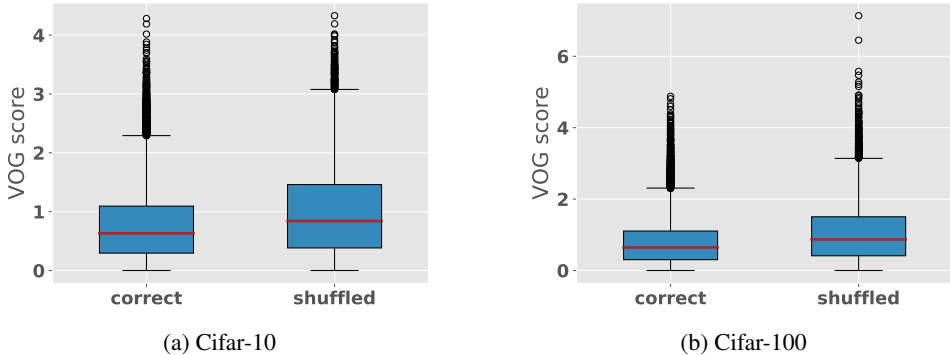

(a) Cifar-10                    (b) Cifar-100

Figure 8: Box-plot of subset the VoG distribution of all examples with *correct* labels against the 20% of the dataset with *shuffled* labels. It is visible that the distribution of VoG scores, both the mean (red line in the plot) and spread, for shuffled data is higher than that of the correct samples for both Cifar-10 (right plot) and Cifar-100 (left plot).

## 5    RELATIONSHIP BETWEEN VoG SCORES AND MEMORIZED/OOD EXAMPLES

Recent work has highlighted that deep neural networks produce output probabilities that are uncalibrated (Guo et al., 2017; Kendall & Gal, 2017; Lakshminarayanan et al., 2017; Hendrycks & Gimpel, 2016) and thus cannot be interpreted as a measure of certainty. If VoG is a useful auditing tool, we expect it to capture model uncertainty even when this is not reflected in the end probabilities.

To this end, we consider VoG rankings on two tasks where the network produces highly certain predictions for incorrect or out-of-distribution inputs.

**Surfacing examples that require memorization** Overparameterized networks have been shown to achieve zero training error by memorizing examples (Zhang et al., 2016; Feldman, 2020). We explore whether VoG is able to identify examples that require memorization and the rest of the dataset. To do this, we replicate the general experiment setup of Zhang et al. (2016) and replace 20% of all labels in

the training set with random shuffled labels. We re-train the model from random initialization and compute VoG scores at relative intervals *across training* for all examples in the training set. Our network achieves $0\%$ training error which would only be possible given successful memorization of the noisy examples with shuffled labels. *Is VoG able to discriminate between these memorized examples and the rest of the dataset?* In Fig. 8, we plot the box plot distribution of VoG scores for the subset of the data with *shuffled* labels that required memorization beside *correct* labels. We find that the mean and spread of the examples with the shuffled labels are higher when compared to the rest of the dataset and that this difference is statistically significant. We perform a two-sample $t$-test with unequal variances. At a p-value of 0.01, we reject the null hypothesis that the mean of both samples are the same and conclude that the difference in VoG means is statistically significant. Shuffled labels have a different VoG distribution than the non-shuffled dataset. We include more details about the statistical testing in the appendix.

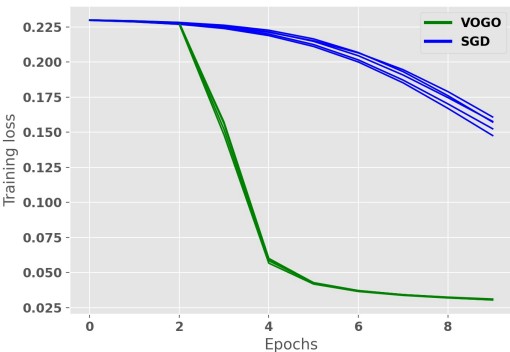

Figure 9: We consider using VoG as a ranking mechanism to accelerate training. VoGo upweights examples using the VoG score during training. Models trained using VoGo (green) converges faster and to a lower training loss as compared to SGD (blue). The VoG scores were calculated using $K = 3$ number of checkpoints and we observe a sharp drop in the training once we start scaling the gradient updates of the batches with their respect VoG scores after the third epoch.

**ImageNet-O experiments** We consider ImageNet-O (Hendrycks et al., 2019), an open source curated out-of-distribution (OOD) dataset designed to fool classifiers. ImageNet-O consists of images that are not included in the original 1000 ImageNet classes. These images were selected with the goal of producing high confidence incorrect ImageNet-1K predictions of labels from within the training distribution. We are interesting in understanding if VoG can correctly rank ImageNet-O examples as being atypical or out of distribution. We would expect to observe that ImageNet-O examples would be over-represented in top percentiles of VoG scores vs the lowest scores. In Fig. 6b, we see that this is indeed the case. We plot the count of ImageNet-O image in each percentile as a fraction of the total count of ImageNet-O images (2000 images in total). ImageNet-O images are relatively over-represented at high levels of VoG, with $30\%$ of all images in the top-25th percentile vs $24\%$ in the bottom 25th percentile.

## 6 CAN WE USE VoG TO ACCELERATE AND IMPROVE TRAINING PERFORMANCE?

In this section, we consider whether the VoG score can be using to improve the optimization process by re-weighting examples considered to be *atypical* examples on the fly during the training process. As a toy experiment, we trained a single layer (32 neurons) feed-forward network on the handwritten MNIST digit classification task. As a preliminary exploration of VoG, we weigh each mini-batch gradient update with their respective VoG scores. We compare this VoGo variant with standard mini-batch Stochastic Gradient Descent (SGD) and Variance Of Gradients Optimizer (VoGo) for 10 epochs using a batch size of 256 (algorithm details of VoGo in the appendix Algorithm 1). As a preliminary implementation of VoGo, we weigh each mini-batch gradient update with their respective VoG scores.

The initial weights of the architectures were set to the same seed for both SGD and VoG optimizers. In Fig. 9, we observe that VoG achieves a lower training loss as compared to SGD. The training difference between the optimizers is reflected in the model's testing performance. Across the 5 different runs, models trained using VoGo achieve a testing accuracy of $91.77 \pm 0.10\%$ as compared to SGD which achieves almost $\approx 20\%$ lower performance at $68.50 \pm 3.09\%$. Notably, the performance deviation across different runs is also smaller for VoGo as compared to its counterpart. Training with VoGo accelerates training, achieves higher test-set accuracy and reduces the stochasticity of the training process.

## 7 RELATED WORK

Our work proposes a method to rank training and test examples by estimated difficulty. Given the size of modern day datasets (Hooker, 2020), this can be a powerful interpretability tool to isolate a tractable subset of examples for human-in-the-loop auditing and also aid in curriculum learning (Bengio et al., 2009). Prior work has proposed different notions of what subset merits surfacing. Early work by (Zhang, 1992; Bien & Tibshirani, 2012; Kim et al., 2015; Kim et al., 2016) that introduced the notion of prototypes, quintessential examples in the dataset, but did not focus on deep neural networks. Kim et al. (2016) also requires assumptions about the statistics of the input distribution. Gal & Ghahramani (2016) showed how we can use dropouts as a Bayesian approximation method for representing model uncertainty in deep learning. Work by Li et al. (2017) requires modifying the architecture to prefix an autoencoder in order to surface a set of prototypes. Koh & Liang (2017) proposes influence functions to identify training points most influential on a given prediction.

Unlike previous works, we propose a measure that can be extended to rank the entire dataset by estimated difficulty (rather than surfacing a prototypical subset). Additionally, ranking individual samples using methods like Koh & Liang (2017) would be extremely computationally extensive. Our method does not require modifying the architecture or making any any assumptions about the statistics of the input distribution. In that sense, our work is complementary to recent work by Jiang et al. (2020) which proposes a c-score to rank each example by alignment with the training instances, Hooker et al. (2019) which classify examples as outliers according to sensitivity to varying model capacity and Carlini et al. (2019) which consider several different measures to isolate prototypes that could conceivably be extended to rank the entire dataset. We note that the c-score method proposed by Jiang et al. (2020) is considerably computationally intensive to compute than VoG as it requires training up to $20,000$ network replications per data set. Several of the propotype methods considered by Carlini et al. (2019) require training ensembles of models, as does the compression sensitivity measure proposed by Hooker et al. (2019). Our method is both different in formulation and can be leveraged using a small number of existing checkpoints saved over the course of training.

## 8 CONCLUSION AND FUTURE WORK

Our methodology offers one way for humans to better understand the relative difficulty of different examples. One of our key findings is that VoG is far more challenging to classify for the algorithm and surfaces clusters of images with distinct visual properties. VoG is straight-forward to compute and can take advantage of current best practices of storing multiple checkpoints over the course of training. In practice, a domain expert may choose to compute VoG for a class of particular interest which would further reduce the computational cost.

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

# A  APPENDIX

**Toy Experiment** We generate the clusters for classification using scikit-learn library. We used a 90-10% split for dividing the dataset into the training and testing category. A linear Multiple Layer Perceptron (MLP) network with a single hidden layer of 10 neurons was trained using Stochastic Gradient Descent (SGD) optimizer for 15 epochs. We divided the training process into three stages: (1) $Early$ [0, 5) epochs, (2) $Middle$ [5, 10), and (3) $Late$ stage [10, 15). Our trained model achieves a 0% testing error using a linear boundary (Fig. 1a).

**Class Level Error Metrics and VoG** Here, we explore whether VoG is able to capture class level differences in difficulty. We compute VoG scores for each image in the test-set of Cifar-10 and Cifar-100 (both test-sets have 10,000 images). In Fig. 10, we plot the average absolute VoG score for each class against the false negative rate for each class. We find that there is a positive, albeit weak, correlation between the two, classes with higher VoG scores have higher mis-classification error rate. The correlation between these metrics is 0.65 and 0.59 for Cifar-10 and Cifar-100 respectively. Given that VoG is computed a per-example level, we find it interesting that the aggregate average of VoG is able to capture class level differences in difficulty.

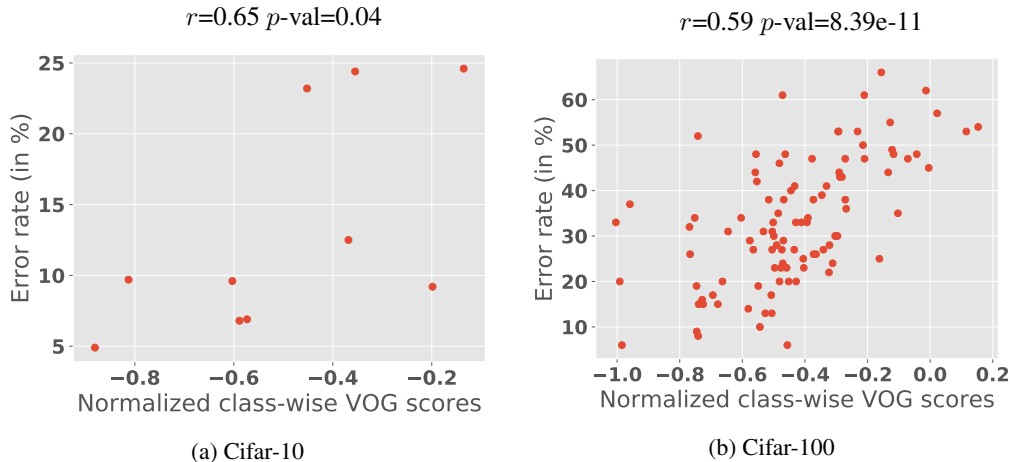

(a) Cifar-10                  (b) Cifar-100

Figure 10: Plot of class false negative rate (y-axis) against average class VoG score for all classes (x-axis). **Left**: Cifar-10 **Right**: Cifar-100. There is a statistically significant positive correlation between class level error metrics and average VoG score (alpha set at 0.05).

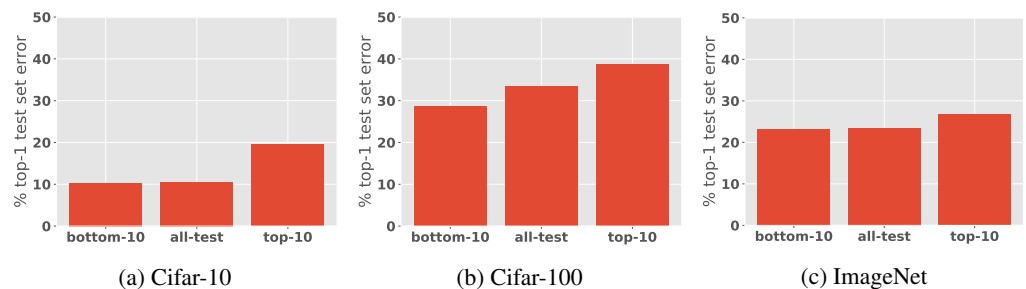

(a) Cifar-10          (b) Cifar-100          (c) ImageNet

Figure 11: Bar plots showing the mean top-1 error rate (in %) for three group of samples from (1) the subset of the test-set with the bottom 10th percentile of VoG scores, (2) the complete testing dataset, and (3) the subset of the test-set with the top 10th percentile of VoG scores.

**Statistical Significance of Memorization Experiments** *Is VoG able to discriminate between these memorized examples and the rest of the dataset?* In Fig. 8, we plot the box plot distribution of VoG scores for the subset of the data with *shuffled* labels that required memorization beside *correct* labels. We find that the mean and spread of the examples with the shuffled labels are higher when compared to the rest of the dataset and that this difference is statistically significant. The two groups

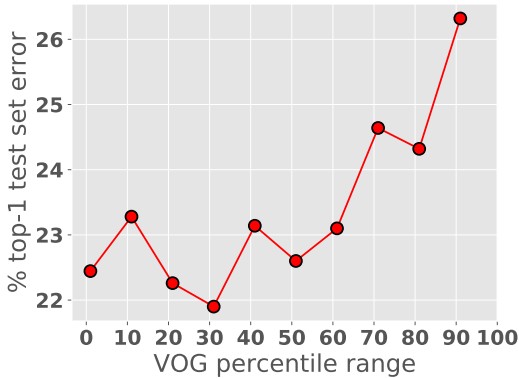

Figure 12: The mean top-1 test set error (y-axis) for the exemplars thresholded by VoG score percentile (x-axis) calculated using the predicted labels. We observe that misclassification increases with an increase in VoG scores. Across ImageNet we observe that VoG calculated for the predicted labels follows the general trend as in Fig. 5 where the top-10 percentile VoG scores have the highest error rate.

of population in the test consists of the VoG scores for the unaltered training set and the subset of the training set with shuffled labels. We conduct this $t$-test for both Cifar-10 and Cifar-100 datasets. The two-sample $t$-test produces a $p$-value that can be used to decide whether there is evidence of a significant difference between the two distributions of VoG scores. The $p$-value represents the probability that the difference between the sample means is large, *i.e.*, the smaller the $p$-value, the stronger is the evidence that the two populations have different means.

Null Hypothesis: $\mu_1 = \mu_2$
Alternative Hypothesis: $\mu_1 \neq \mu_2$

If the $p$-value is less than your significance level ($\alpha = 0.05$ in this experiment), you can reject the null hypothesis, *i.e.*, the difference between the two means is statistically significant. The details for the individual $t$-tests for Cifar-10 and Cifar-100 are given below:

**Cifar-10:** The statistics for the samples in the correct and shuffled labels are:
Corrected labels: $\mu_1 = 0.62; \sigma_1 = 0.54; N_1 = 40000$
Shuffled labels: $\mu_2 = 0.85; \sigma_2 = 0.75; N_2 = 10000$
$p$-value is $< 0.001$
Result: Reject Null Hypothesis (the two populations have different VoG means)

**Cifar-100:** The statistics for the samples in the correct and shuffled labels are:
Corrected labels: $\mu_1 = 0.54; \sigma_1 = 0.46; N_1 = 40000$
Shuffled labels: $\mu_2 = 0.82; \sigma_2 = 0.71; N_2 = 10000$
$p$-value is $< 0.001$
Result: Reject Null Hypothesis (the two populations have different VoG means)

## B    VARIANCE OF GRADIENTS OPTIMIZER (VOGO)

Let us consider a deep neural network $F$ that is to be trained on a supervised classification problem. The function $F$ maps an input variable $X$ to an output variable $Y$, formally $F : X \mapsto Y$. Each $y \in Y$ corresponds to one of $C$ categories or classes. We compute the gradient $S$ of the activation $A_p^l$ with respect to each input in the training set. As before, $l$ can represent the pre- or post-softmax layer of the network and $p$ is the index of either the true or the predicted class probability.

$$\mathbf{S} = \frac{\partial A_p^l}{\partial X}$$

In Algorithm 1, we provide a brief overview of the Variance Of Gradients Optimizer (VoGo). For simplicity, we have not mentioned the update steps for the bias parameters. As an initial test, we weight each mini-batch gradient update with their respective VoG scores. Batches having a higher

VoG score are relatively harder for the model to run and hence we up weight their updates in the direction of the gradient descent.

---

**Algorithm 1** Variance of Gradient optimizer

---

1: **Initialization:** Choose an initial value of $\mathbf{w}$; step size $\gamma$ for storing the number of snapshots (set to 3 for the MNIST experiment); learning rate $\eta$; VoG regularization coefficient $\mu$.
2: **for** t < total epochs **do**
3:     $S_t$ = dict()
4:     **for** i, $(\mathbf{x}, \mathbf{y})$ in enumerate($\mathbf{X}, \mathbf{Y}$) **do**
5:         $vog_i = 1$
6:         $y_{pred} = F(x)$
7:         $L = \text{Loss}(y_{pred}, y)$
8:         $S_{temp} = \frac{\partial A_p^l}{\partial x}$
9:         **if** len($S_t$) < $\gamma$ **then**
10:             $S_t[i].append(S_{temp})$
11:         **else**
12:             $S_t[i] = S_t[i][1:] + [S_{temp}]$
13:         **end if**
14:         **if** $t \geq \gamma - 1$ **then**
15:             $vog_i = \text{VOG}(S_t[i])$
16:         **end if**
17:         $\mathbf{w_{t+1}} = \mathbf{w_t} - \mu \times vog_i \times \eta \nabla L(\mathbf{w_t})$
18:     **end for**
19: **end for**

---

