# OpenReview forum: "Estimating Example Difficulty using Variance of Gradients"
_ICLR.cc/2021/Conference — Reject_

### Official Review · AnonReviewer4 · 2020-10-29
**Nicely written paper that does not include a comparison with baselines/other methods**

**Rating:** 3
**Confidence:** 3

**Review:**

The paper proposes a new metric for measuring example difficulty: variance of gradients. Given a neural network for image classification, it measures the variability in the gradient of the output of the penultimate layer of the network with respect to each pixel, as observed during the training process. This quantity is averaged over all pixels to measure the difficulty of an image. Illustrative examples, observed correlation with error rate on test data, and successful identification of out-of-distribution examples indicate that the proposed metric is useful.

The paper is very nicely written and a pleasure to read.

The primary shortcoming of the paper is that it does not compare to any other measure of example difficulty. An obvious metric is the entropy of the estimated class probability distribution for an example (computed either with or without some form of calibration of the probability estimates). It is unclear whether the proposed metric provides a significant amount of additional information.

Although intuitively plausible, and backed up by experimental results, the metric is not justified by providing some form of theory.

Data augmentation is used in the experiments but the paper does not explain how this is accounted for in the calculation of the metric. Is the metric computed for the unmodified training examples only?

Figure 6 (left): Why not compute the consistency of the rankings directly? The difference in observed error is an indirect measure.

Typos:

"p is the index of either the true or the predicted class probability."

"ImaageNet-O"

---

> ### Author Response · Authors · 2020-11-22
> **Response to R4**
>
> We thank R4 for the valuable feedback and questions. We take this opportunity to clarify the contributions of our work and the individual questions below.
>
> ### Data augmentation is used in the experiments but the paper does not explain how this is accounted for in the calculation of the metric. Is the metric computed for the unmodified training examples only?
> In this work, we wanted to evaluate the effectiveness of the VoG score in a classical DNN training regime where the networks are trained using a combination of the real and augmented samples. The metric was calculated without any modification to the original training examples. However, we agree that we can also rank the real and augmented samples individually using the VoG scores.
>
> ### Figure 6 (left): Why not compute the consistency of the rankings directly? The difference in observed error is an indirect measure.
> In this experiment, we wanted to evaluate the effectiveness of the score in a downstream task. Comparing the direct distribution of the VoG score can be misleading as VoG scores from different runs follow a normal distribution (verified through experiments) but they do not highlight how interpretable those scores are. Hence, we compared the stability using the misclassification error performance.
>
> Thank you for your valuable feedback and we plan to include the other suggested changes.

---

### Official Review · AnonReviewer3 · 2020-10-31
**An appealing idea for difficulty estimation, but requires more thorough comparison to prior methods.**

**Rating:** 4
**Confidence:** 4

**Review:**

Summary

The authors propose *variance-of-gradients* (VOG) as a metric for assessing the difficulty of an example for a model.  For each example, the gradients of the model loss are backpropagated to the input features (pixels for images) for a sample of checkpoints, the per-pixel standard deviation is calculated over the checkpoints, and then the standard deviation is averaged over pixels. The authors present evidence that the VOG metric can differentiate difficult and easy examples, both for labeled examples, and also for unlabeled examples, where a label is imputed as the model prediction. The metric also discriminates visually distinct image types, and shows some correlation with examples that had to be memorized by the model.

Pros:

- VOG is an intuitively appealing metric.  If the gradients for an example vary wildly over training, it would seem to be an indicator that the model's representation of that example is unstable, and conflicting signals are present in the example.  Exploration of gradient statistics is a fruitful area for understanding model behavior and training dynamics.

- The authors examine VOG from several interesting viewpoints.  It can be used as a trustworthiness indicator (correlates with accuracy), but also seems to correlate with out-of-distribution examples and example memorization.

- The correlation pattern with accuracy reverses from early- to late-stage training, which gives an interesting viewpoint into training dynamics: early on, gradient updates have high variance for easy examples, and later on, gradient variance is high for the examples that remain difficult for the classifier.

Cons:

Overall, while VOG clearly shows some useful properties, there could be more clear benchmarking and comparison to other methods for detection of example difficulty.  The paper currently reads as thorough data analysis showing intriguing results, but without yet clear enough metrics to judge VOG in the context of the many methods for model trustworthiness / OOD detection.  The paper does cite and describe relevant prior research, but more quantitative comparison is called for.

One of the most important figures is 6b, wherein it is shown that out-of-distribution data is concentrated at the higher quantiles of VOG.  However, the authors should present some baselines here:
*    The actual probability of the highest confidence class.  The authors state that "DNNs produce output probabilities that are uncalibrated and thus cannot be interpreted as a measure of certainty".  While it true that the output probabilities can be problematic, I would assert that they are still widely used, and there is some onus on the authors to demonstrate the superiority of VOG empirically.
*   There are numerous methods for classify-with-abstain, and one-class classification which could be benchmarked against.  Such methods have been extended to neural architectures.  A few examples that come to mind:
    *    Liu, Z., Wang, Z., Liang, P. P., Salakhutdinov, R. R., Morency, L. P., & Ueda, M. (2019). Deep gamblers: Learning to abstain with portfolio theory. In Advances in Neural Information Processing Systems (pp. 10623-10633).
    *    Jiang, H., Kim, B., Guan, M., & Gupta, M. (2018). To trust or not to trust a classifier. In Advances in neural information processing systems (pp. 5541-5552).
    *    Malinin, A., & Gales, M. (2018). Predictive uncertainty estimation via prior networks. In Advances in Neural Information Processing Systems (pp. 7047-7058).
    *    One-class classification is related, and could be used to detect out-of-distribution data.  Ruff, L., Vandermeulen, R., Goernitz, N., Deecke, L., Siddiqui, S. A., Binder, A., ... & Kloft, M. (2018, July). Deep one-class classification. In International conference on machine learning (pp. 4393-4402).
    *    Hendrycks, D., Mazeika, M., & Dietterich, T. (2018). Deep anomaly detection with outlier exposure. arXiv preprint arXiv:1812.04606.

Of course, not all of these methods necessarily need to be benchmarked, but the most representative or competitive methods should have comparisons.

Overall the VOG distributions show quite a bit of overlap between in and out-of-distribution examples on ImageNet-O, and correct and shuffled examples on the Cifar-10 and Cifar-100 datasets.  Hence it is not a 100% reliable indicator and needs to be more firmly placed in context.

A few specific comments:
*   In the "Contributions" section, the authors state "Restricting evaluation to the test-set examples with the lowest VOG greatly improves generalization performance”.  This confused me, as the authors show that lowest VOG examples have higher accuracy; however this seems to be different from generalization performance being improved, which is a statement about the model quality.
*    VOG shows vastly different dynamics early and late in training, which is an interesting result.  Hence, in practice, should we use late-VOG for difficulty estimation?  The authors should present clearer guidance here.
*    In Figure 6a, the authors show that the relationship between VOG and test set error is stable between model retrains.  However, I would argue that it would be a much stronger result to show that VOG is a stable property of the image, as this would show example difficulty for a model class.

A few proof-reading issues:
*    In "Methodology", the authors reference $A^l_n$ without defining $A$.  From the context it seems this is activation, but it causes a bit of a confusion when reading.
*    Equation (2) contains a subscript $i$ on the left-hand-side, whereas the right hand side sums over $i$.  I suspect the LHS should have a distinct index for the example.
*   Hendrycks et al., "Natural adversarial examples", has two different entries in "References".

---

> ### Author Response · Authors · 2020-11-19
> **Response to R3**
>
> We thank R3 for the valuable feedback and questions. We take this opportunity to clarify the contributions of our work and the individual questions below.
>
> ### Overall the VOG distributions show quite a bit of overlap between in and out-of-distribution examples on ImageNet-O, and correct and shuffled examples on the Cifar-10 and Cifar-100 datasets. Hence it is not a 100% reliable indicator and needs to be more firmly placed in context.
>
> We agree with R3’s observation that there is an overlap between the two distributions when we compare correct and shuffled or ImageNet and ImageNet-O examples. We would like to clarify that the high VoG samples do not guarantee that the model incorrectly classifies the image in question. However, we note that this does not preclude the ranking from correctly surfacing examples of what is most challenging. For example, in the case where VoG is computed for the training set, it is possible the model learns to memorize the correct label for a difficult example but this example is still more relatively more challenging than a far easier example which is learnt early in training. We do in fact see this to be the case in our label shuffling experiment (Sec. 5). Here, the model still achieves 0% training error despite the shuffled labels, but VoG ranks the shuffled labels as higher than the other labels.
>
> In light of the question of whether the results are significant and reliable, we provide evidence that the means of the two populations of VoG scores in the memorization experiments are significantly different. We perform a two-sample t-test with unequal variances. The two groups of population in the test consists of the VoG scores for the unaltered training set and the subset of the training set with shuffled labels. We conduct this t-test for both CIFAR-10 and CIFAR-100 datasets. The two-sample t-test produces a p-value which can be used to decide whether there is evidence of a significant difference between the two distributions of VoG scores. The p-value represents the probability that the difference between the sample means is large, i.e., the smaller the p-value, the stronger is the evidence that the two populations have different means.
>
>
> Null Hypothesis: $\mu_{1}=\mu_{2}$
>
> Alternative Hypothesis: $\mu_{1} \neq \mu_{2}$
>
> If the $p$-value is less than your significance level ($\alpha = 0.05$ in this experiment ), you can reject the null hypothesis, i.e., the difference between the two means is statistically significant.
> The details for the individual t-tests for C10 and C100 are given below:
>
>
> CIFAR-10
>
> Corrected labels → $\mu_{1}=0.62$; $\sigma_{1}=0.54$; $N_{1}=40000$
>
> Shuffled labels → $\mu_{2}=0.85$; $\sigma_{2}=0.75$; $N_{2}=10000$
>
> P-value is $<0.001$
>
> Result: Reject Null Hypothesis (the two populations have different VoG means)
>
>
> CIFAR-100
>
> Corrected labels → $\mu_{1}=0.54$; $\sigma_{1}=0.46$; $N_{1}=40000$
>
> Shuffled labels → $\mu_{2}=0.82$; $\sigma_{2}=0.71$; $N_{2}=10000$
>
> P-value is $<0.001$
>
> Result: Reject Null Hypothesis (the two populations have different VoG means)
>
> Summary of conclusions: For both Cifar-10 and Cifar-100, we reject the Null Hypothesis and conclude that the difference in VoG means is statistically significant. Shuffled labels have a different VoG distribution than the non-shuffled dataset.
>
> ### VOG shows vastly different dynamics early and late in training, which is an interesting result. Hence, in practice, should we use late-VOG for difficulty estimation? The authors should present clearer guidance here.
> Yes, in practice, we should use late-VOG for difficulty estimation. The early-stage dynamics were just shown to analyze the contrast in the learning process of the model.
>
> ### In "Methodology", the authors reference without defining. From the context it seems this is activation, but it causes a bit of a confusion when reading.
> Thank you for catching the typo. It should be $A_{p}^{l}$ which is the activation at the pre-softmax layer $l$ of the network and $p$ is the index of either the true or predicted class probability. We will make the respective changes in the final version and detail the formulation of the VoG score.
> We will fix the other proof-reading issues in the final manuscript.
>
> Based on the R3’s feedback, we will update our manuscript with the above test results.

---

### Official Review · AnonReviewer1 · 2020-10-31
**Computing variance of gradient for estimating example difficulty during neural network training**

**Rating:** 6
**Confidence:** 4

**Review:**

##########################################################################

Summary:

The authors propose to use the variance of the gradient (VOG) values measured during neural network training as an estimate of the difficulty of examples. For each example, the gradient of the loss function is computed with respect to each pixel during training. The VOG is the mean over all the pixels of the variance of the gradient over different training steps. The author show that VOG is correlated to train and test set error (highest VOG sample have highest error rate). VOG also shows that easy example (with lower error rate) are leant first during training. Out of distribution sample or sample with incorrect label also show a higher VOG mean and a larger variance. VOG can therefore be used to detect incorrectly labeled samples.


##########################################################################

Reasons for score:


The idea proposed in this paper is simple and seems to be effective. However, the authors should demonstrate that in practice VOG can be used to increase the accuracy :
* is it possible to use VOG as a confidence score and classify samples only when their VOG is above a given threshold ? at threshold 0, this is the classical setup. At threshold 0.8 (if voG can be normalized between 0 and 1), only sample with VOG >0.8 are classified. This should lead to higher accuracy. Is it the case ? To what extend ?
* what is the computational complexity of computing VOG on test set to detect outliers or difficult examples ? How many training step s on test samples have to be done ?


Page 3, A^l_n is not defined ;  in the equation, S should be S_{ti}

The gradient is computed over all the pixel of the image. If the image is large, this could be too computational expensive. Is it possible to sample the pixels and still have a good estimate of the VOG ?


Page 4, Figure 3 : the sample generated by data augmentation seem to be the most difficult sample. The same inspection should be conducted on the original samples only. On the other hand, could VOG be used to guide the data augmentation process by selecting the "difficult" samples ?


Page 5, figure 4. The y axis is different for the different figure, which is misleading.
Page 6, figure 6 a : add the legend, it is not obvious at the first sight that there are 5 curves.
Page 7, figure 7 : could you explain how is computed VOG on test set sample ? are they added to the training set without updating the gradient ? for the curves, is the VOG computed on all the training steps or only on the same number of steps at the beginning or end of the training ?

Related works : papers on curriculum learning  and bayesian approximation could be cited
- Yoshua Bengio, Jérôme Louradour, Ronan Collobert, and Jason Weston. 2009. Curriculum learning. In Proceedings of the 26th Annual International Conference on Machine Learning (ICML '09). Association for Computing Machinery, New York, NY, USA, 41–48. DOI:https://doi.org/10.1145/1553374.1553380
- Yarin Gal and Zoubin Ghahramani. 2016. Dropout as a Bayesian approximation: representing model uncertainty in deep learning. In Proceedings of the 33rd International Conference on International Conference on Machine Learning - Volume 48 (ICML'16). JMLR.org, 1050–1059.

---

> ### Author Response · Authors · 2020-11-16
> **Response to R1: Part 1**
>
> We thank R1 for the valuable feedback and questions. We take this opportunity to clarify the contributions of our work and individual questions below.
>
> ### However, the authors should demonstrate that in practice VOG can be used to increase the accuracy
>
> 1. Can VoG be used to increase the training accuracy?
> In light of R1’s suggestion, we examined the effects of leveraging the VoG score to facilitate the training of certain "atypical" examples on the fly during the training process. As a toy experiment, we trained a single layer (32 neurons) feed-forward network on the handwritten MNIST digit classification task. For comparison, we trained the model with both Stochastic Gradient Descent (SGD) and Variance Of Gradients Optimizer (VOGO) for 10 epochs using a batch size of 256. As a preliminary implementation of VOGO, we weigh each mini-batch gradient update with their respective VoG scores. Batches having a higher VoG score are relatively harder for the model to run and hence we up weight their updates in the direction of the gradient descent. The initial weights of the architectures were set to the same seed for both SGD and VoG optimizers. In Fig. 1, we observe that the VOGO optimizer achieves a lower training loss as compared to SGD. The training difference between the optimizers is reflected in the model's testing performance. Across the 5 different runs, models trained using VOGO achieve a testing accuracy of $91.77\pm0.10$% as compared to SGD which achieves almost $\approx20$% lower performance at $68.50\pm3.09$%. Notably, the performance deviation across different runs is also smaller for VOGO as compared to its counterpart.
> 2. what is the computational complexity of computing VOG on a test set to detect outliers or difficult examples? How many training steps on test samples have to be done?
> The VoG scores just use the backpropagated gradients with respect to the input image. The computational complexity is directly proportional to the number of checkpoints and the size of the input image. In both our VoG and VOGO experiments, we used it is possible to compute an informative ranking using as few as 3 checkpoints for calculating the VoG score.
>
> ### Page 3, A^l_n is not defined ; in the equation, S should be S_{ti}
>
> $A_{p}^{l}$ is the activation at the pre-softmax layer $l$ of the network and $p$ is the index of either the true or predicted class probability. We will make the respective changes in the final version and detail the formulation of the VoG score.
>
> ### Is it possible to sample the pixels and still have a good estimate of the VOG?
>
> With respect to computational complexity, we would like to clarify that our proposed VoG metric is calculated only using different snapshots over the course of training. This allows it to fit into a typical training regime sitting. Other proposed confidence estimation approaches either did not focus on DNNs (Zhang, 1992; Bien & Tibshirani, 2012; Kim et al., 2015; Kim et al., 2016) or required architectural modification (Li et al. (2017)) or were computationally expensive (Koh & Liang (2017), Jiang et al. (2020)). Unlike these measures, our proposed VoG metric does not require any additional training or architectural modification and just uses the intermediate weight checkpoints which are often saved for analyzing the training process. Our method is both different in the formulation and can be leveraged using a small number of existing checkpoints saved over the course of training whereas methods like the c-score (Jiang et al. (2020)) is considerably computationally intensive to compute as it requires training up to 20,000 network replications per data set.
> We agree with the reviewer that the idea of sampling pixels and approximating VoG is an interesting one worthy of future exploration.
>
> ### the sample generated by data augmentation seem to be the most difficult sample. The same inspection should be conducted on the original samples only. could VOG be used to guide the data augmentation process by selecting the "difficult" samples?
>
> In this work, we wanted to evaluate the effectiveness of the VoG score in a classical DNN training regime where the networks are trained using a combination of the real and augmented samples but we agree that we can rank the real and augmented samples individually using the VoG scores too. Our results from VOGO (described in comment 1) highlight that we can leverage the VoG score and guide the training of the network by attending to more difficult samples (which can the augmented samples in this case).

---

> ### Author Response · Authors · 2020-11-16
> **Response to R1: Part 2**
>
> We thank R1 for the valuable feedback and questions. We take this opportunity to clarify the contributions of our work and individual questions below.
>
> ### could you explain how is computed VOG on test set sample?
>
> Thank you for the pointers to the figures. The y-axis of Fig. 4 was kept different so as to highlight the behavior of VoG on individual datasets. In light of your comment, we have added the legends in Fig. 7. Regarding the test samples, we did not use them during the training cycle. For calculating the VoG for the testing samples, we load the weights of the model at individual checkpoints and generate the respective gradients for the test samples post-training. Once we have the gradients, we follow Eqn. (2) (Sec. 2) and calculate their respective VoG scores.
>
> ### Related works: papers on curriculum learning and bayesian approximation could be cited
>
> Thank you for sharing the papers. We would be very happy to add the citations and will make sure it is reflected in the final manuscript.

---

### Official Review · AnonReviewer2 · 2020-11-02
**Powerful tool, experimental results need some more work**

**Rating:** 6
**Confidence:** 4

**Review:**

Summary:

The authors propose Variance of Gradients (VoG) as a quantifiable metric to identify examples that are difficult to classify. This is motivated by the intuition that examples that are easy to classify do not contribute much to the loss beyond early stages of training, hence don't contribute much to the gradient. The gradient $S_{it}$ w.r.t. every pixel $i$ is tracked across $t=1..K$ snapshots through the training process. The mean $\mu_i$ of $S_{it}$ is computed across snapshots. and finally the VOG of each example $j$ is computed as the average over all $N$ pixels and $K$ snapshots of the squared difference between the gradient $S_{it}$ and $\mu_i$.
Qualitative and quantitative proof of correlation of VoG with diffuculty of examples is provided:
Qualitative: Authors illustrate that examples with high VoG tend to have cluttered backgrounds or odd angles.
Quantitative: High correlation betwee number of erroneous examples in test set and associated VoG, especially on the harder datasets like ImageNet.
VoG is used to study phases of training a neural network. Interestingly, if we focus only on early epochs of training, the difficult examples have low VoG while easy examples have high VoG. As in early phases of training, models concentrate on learning the easy examples, and the error rate on these easy examples is still high.
VoG is also used to identify out of distribution examples that high capacity models tend to get right only by memorization.

Strengths:
- A new metric, VoG, is proposed that is easy to calculate and shown to be associated with examples that are difficult to classify. Having such a powerful tool will be useful for a variety of purposes, from identifying atypical examples for human auditing, to aided interpretability of models.
- The metric is easy to compute as compared to competing methods. It can easily be adopted by practitioners as they using checkpoints that are often computed and saved anyway.
- Empirical results are convincing for the most part. There is a clear increase in error-rate as VoG increases. And this relationship is shown to hold across various network initializations. VoG is shown to vary throughout training. As the network begins to converge, the difficult examples show consistently high VoG.

Weaknesses:
- Fig. 4 is not clear. For each value of decile on the x-axis, the error rate is computed on that 10% of data. And the error rate is shown to be higher for larger values of VoG. However, the maximum error rate even for the maximum value of VoG is 20-40%. Therefore, there are clearly upto 80% of examples with high VoG that are still correctly classified. Authors don't explain why this is the case.
- Fig. 7: It is not clear why the error rate associated with the difficult examples that have low VoG early in training is low. Shouldn't the errors associated with difficult examples remain the same throughout training i.e., the network never learns to classify these examples correctly. Why would the error rate degrade? Or are the authors reporting percentage of total errors on the y-axis. In which case while the asbolute number of errors associated with difficult examples remains the same, their relative ratio as compared to the overall number of errors increases as training proceeds. Please clarify.
- Fig. 8: Out of distribution examples e.g., deliberately shuffled labels are shown to be associated with slightly higher VoG score values. Can the authors include a significance test to show this is a material difference. There is a high variance in VoG values for shuffled examples. Why would these examples exhibit lower VoG? Can the authors provide some intuition behind this.

Conclusion:
Overall, my decision is to accept the paper because this is a powerful proposal that deserves to be investigated further. However, I have some reservations about the empirical results as described above. If authors can explain/clarify these aspects it would be a much stronger submission.

In addition to those listed above, the authors should address the questions below in future work:
- How come not all or even a majority of examples that are misclassified by a network have high VoG?
- Will results hold across various types of models? What is the relationship of VoG with capacity of models?
- Will results hold across domains?

---

> ### Author Response · Authors · 2020-11-16
> **Response to R2**
>
> We thank R2 for the valuable feedback and questions. We take this opportunity to clarify the contributions of our work and individual questions below.
>
> ### Fig. 4 is not clear.
>
> We agree with R2’s observation that high VoG samples do not guarantee that the model incorrectly classifies the image in question. However, we note that this does not preclude the ranking from correctly surfacing what is most challenging. For example, in the case where VoG is computed for the training set, it is possible the model learns to memorize the correct label for a difficult example but this example is still more relatively more challenging than a far easier example which is learnt early in training. We do in fact see this to be the case in our label shuffling experiment (Sec. 5). Here, the model still achieves 0% training error despite the shuffled labels, but VoG ranks the shuffled labels as higher than the other labels.
>
> ### Fig. 7: It is not clear why the error rate associated with the difficult examples that have low VoG early in training is low.
>
> In Fig. 7 we show how the behavior of VoG flip as the training progresses from the early to the late stage. This observation is in line with recent works that show that there are distinct stages to training in deep neural networks (Achille et al., 2017; Jiang et al., 2020; Mangalam & Prabhu, 2019; Faghri et al., 2020) where the model first learns easy examples and then proceeds to learn, relatively, harder examples. Building on these works, we show that the higher VoG examples in the early-stage have a lower error rate (Fig. 7a) as the gradient updates center around easy examples. However, in the late-stage when the model proceeds to learn harder examples the trend is flipped as the gradients for the easy samples, by this stage, have already converged (lower VoG) and the harder examples dominate the computation of variance (Fig. 7b). Thus, examples having higher VoG at the end of the training process are misclassified more.
>
> ### Fig. 8: Out of distribution examples e.g., deliberately shuffled labels are shown to be associated with slightly higher VoG score values. Can the authors include a significance test to show this is a material difference.
>
> In light of R2's comment and to provide evidence that the means of the two populations of VoG scores in the memorization experiments are significantly different, we perform a two-sample t-test with unequal variances. The two groups of population in the test consists of the VoG scores for the unaltered training set and the subset of the training set with shuffled labels. We conduct this t-test for both Cifar-10 and Cifar-100 datasets. The two-sample t-test produces a p-value that can be used to decide whether there is evidence of a significant difference between the two distributions of VoG scores. The p-value represents the probability that the difference between the sample means is large, i.e., the smaller the p-value, the stronger is the evidence that the two populations have different means.
>
> Null Hypothesis: $\mu_{1}=\mu_{2}$
>
> Alternative Hypothesis: $\mu_{1} \neq \mu_{2}$
>
>
> If the p-value is less than your significance level ($\alpha = 0.05$ in this experiment), you can reject the null hypothesis, i.e., the difference between the two means is statistically significant. The details for the individual t-tests for C10 and C100 are given below:
>
> Cifar-10
>
> Corrected labels → $\mu_{1}=0.62$; $\sigma_{1}=0.54$; $N_{1}=40000$
>
> Shuffled labels → $\mu_{2}=0.85$; $\sigma_{2}=0.75$; $N_{2}=10000$
>
> P-value is $<0.001$
>
> Result: Reject Null Hypothesis (the two populations have different VoG means)
>
>
>
> Cifar-100
>
> Corrected labels → $\mu_{1}=0.54$; $\sigma_{1}=0.46$; $N_{1}=40000$
>
> Shuffled labels → $\mu_{2}=0.82$; $\sigma_{2}=0.71$; $N_{2}=10000$
>
> P-value is $<0.001$
>
> Result: Reject Null Hypothesis (the two populations have different VoG means)
>
>
> Summary of conclusions: For both Cifar-10 and Cifar-100, we reject the Null Hypothesis and conclude that the difference in VoG means is statistically significant. Shuffled labels have a different VoG distribution than the non-shuffled dataset.
>
> ### Will results hold across various types of models? What is the relationship of VoG with capacity of models?
>
> Yes, the VoG results hold across models with different capacities. In our work, we presented the results across 2 architectures and 3 datasets where the capacity of the two networks was widely different, i.e. ResNet-18 has 11M parameters whereas ResNet-50 has 23M parameters.
>
> ### Will results hold across domains?
> VoG is domain-independent as it only uses the backpropagated gradient across different checkpoints. Hence, for a given input, we can calculate the VoG score for any differentiable model.
>
> Based on the R2's feedback, we will update our manuscript to better clarify the points made above.

---

### Official Review · AnonReviewer5 · 2020-11-06
**The authors propose a simple, but effective technique for discovering challenging examples.**

**Rating:** 6
**Confidence:** 4

**Review:**

The authors propose to use a scalar measures called Variance of Gradients to discover challenging-to-learn examples on which the model is more likely to make an error. They illustrate that low VoG examples are "prototypical" and more easily understood, while high VoG examples typical exhibit occlusions, strange angles, zooms, cluttered backgrounds and other characteristics which make they visually also challenging to classify. Furthermore, high VoG examples are also the ones which the model finds more challenging to correctly classify.

In terms of dataset interpretability and analysis, I think this work provides interesting empirical insights, especially using such a simple method, which is always good.

The work is clearly written (with a notable exception discussed below), easy to read, conceptually straightforward and information. However, this work has a number of issues, which (at the moment) prevent me from recommending acceptance.

1. The authors do not actually provide a clear definition of VoG - it is not clear WHAT we are taking the gradient off with respect to an input pixel - the predicted class probability? S is a matrix, but the average of S (mu_i) is a scalar. The notation is opaque and needs further clarification. Furthermore, it is not clear how class normalisation by mean-std is done. Given that the VoG is a central aspect of this work, it MUST be clarified to remove any and all ambiguities. (NECESSARY)

2 In their synthetic analysis the authors show that high VoG example typically lie on the decision boundary between classes and therefore exhibit high data uncertainty (or high aleatoric uncertainty). In practice, judging from the picture, this seems to be also the case in practice on C10/C100/Imagenet. However, the authors do not answer *WHY* examples on decision boundaries should have a high VoG score. (Theoretical interpretation necessary)

3. While I understand that this is an empirical work, I think it requires discussion of the relationships of the proposed measure to other uncertainty/difficulty/confidence estimation approaches is necessary. Of particular importance is the relationship to ensembles of models. Note note the VoG,  which examines the average per-pixel variance of the gradient across an ensemble of models from different checkpoints (stages of training), relates to other measures of ensemble diversity. (Discussion necessary)

Finally, it would be interesting if the authors examined the effects of filtering out training data based on VoG. (Nice to have, but not necessary).

---

> ### Author Response · Authors · 2020-11-15
> **Response to R5: Part 1**
>
> We thank R5 for the valuable feedback and questions. We take this opportunity to clarify the contributions of our work and individual questions below.
> ### provide a clear definition of VoG
> Let us consider a supervised classification problem where a DNN is trained to approximate the function $F$ that maps an input variable $X$ to an output variable $Y$, formally $F: X \mapsto Y$.
> $y \in Y$ is a discrete label vector associated with each input $X$. Each label y corresponds to one of $C$ categories. A given input image $X$ can be decomposed into a set of pixels $x_i$, where $i=${$1, \dots, N$} and $N$ is the total number of pixels in the image. For a given image, we compute the gradient of the activation $A_{p}^{l}$ with respect to each pixel $x_i$. Here, $l$ designates the pre-softmax layer of the network and $p$ is the index of either the true or predicted class probability. We consider $S$ as a matrix that represents the gradient of $A_{p}^{l}$ with respect to individual pixels $x_i$, i.e., for an image of (say) $3\times32\times32$ size, $S$ will be a  $3\times32\times32$ gradient matrix.
> $$\mathbf{S} = \frac{\partial A_p^{l}}{\partial x_i}$$
> Without loss of generality, we take the sum across the color channels to arrive at a gradient matrix $S$ where $S \in \mathbb{R}^{32\times32}$. For a given set of $K$ checkpoints, we generate the above gradient matrix $S$ for all individual checkpoints, i.e., {$S_{1}, \dots, S_{K}$}. We then calculate the mean gradient matrix $\mu$ by taking the average of the $K$ gradient matrices. Note, $\mu$ is the mean across different checkpoints and is of the same size as the gradient matrix $S$. We then calculate the variance of gradients across each pixel using the equation $$VoG_{pixel} = \sqrt \frac{1}{K}\sum_{t=1}^{K}(S_{t} -\mu)^{2}$$
> Here, $VoG_{pixel}$ is a matrix representing the variance of gradients of each pixel in the image. We average the pixel-wise variance of gradients to compute a scalar VoG score for the given input image: $$VoG=\frac{1}{N}sum(VoG_{pixel})$$
> To calculate the class-normalized VoG score, we calculate the mean and deviation of all the VoG scores belonging to each class $c$, where $c \in$ {$1, \dots C$} from the training dataset and then normalize the VoG scores of individual images by their respective class mean and deviation.
>
> We thank the reviewer for the suggestions to detail VoG in more clarity, and we will update the manuscript accordingly.
> ### Discussion of the relationships of the proposed measure to other uncertainty/difficulty/confidence estimation approaches and Of particular importance is the relationship to ensembles of models.
> We thank the reviewer for directing us in the direction of leveraging an ensemble of models for ranking samples. We believe this is an interesting direction for future work where, rather than comparing the checkpoints over the course of training of a single model, we compare across similar training stage checkpoints across different models.
> In essence, we would like to clarify that our proposed VoG metric is not calculated using an ensemble of models, but rather computed using different snapshots of the same model over the course of training. This allows it to fit into a typical training regime sitting. Other proposed confidence estimation approaches either did not focus on deep neural networks (Zhang, 1992; Bien & Tibshirani, 2012; Kim et al., 2015; Kim et al., 2016) or required architectural modification (Li et al. (2017)) or were computationally expensive (Koh & Liang (2017), Jiang et al. (2020)). Unlike these measures, our proposed VoG metric does not require any additional training or modification and just uses the intermediate weight checkpoints which are often saved for analyzing the training process. Our method is both different in the formulation and can be leveraged using a small number of existing checkpoints saved over the course of training whereas methods like the c-score (Jiang et al. (2020)) is considerably computationally intensive to compute as it requires training up to 20,000 network replications per data set.

---

> > ### Comment · AnonReviewer5 · 2020-11-24
> > **Response**
> >
> > Thanks for your effort!
> >
> > The definition is now much clearer. The additional data filtering experiment also sounds sensible, as far as I can tell. However, I would still like to see the changes reflected in the manuscript. As far as I can tell, it has not been modified yet.
> >
> > Regards,
> > R5

---

> > > ### Author Response · Authors · 2020-11-24
> > > **Response to R5**
> > >
> > > Thank you for your comment, R5. Apologies for the oversight, we were uncertain of the appropriate timeline of when to update the changes. Thank you for the clarification. We have gone ahead and included the new detailed formulation, added in the details of the statistical tests for the memorization experiments, and also the experiment details for analyzing the effects of filtering out training data based on VoG in the rebuttal version of the manuscript. This last experiment also includes the charts which illustrate the acceleration in training convergence achieved by switching to VoG as an optimizer, as suggested by R5 during the initial reviews.

---

> > > > ### Comment · AnonReviewer5 · 2020-11-24
> > > > **Thanks for your effort!**
> > > >
> > > > Thank you, you have addressed my main concerns. I will increase the score, though my final score will be decided during the final phase of the rebuttal process.
> > > >
> > > > Sincerely,
> > > > R5

---

> ### Author Response · Authors · 2020-11-15
> **Response to R5: Part 2**
>
> ### Finally, it would be interesting if the authors examined the effects of filtering out training data based on VoG.
>
> This is an interesting idea. During the rebuttal period, we went forward with R5 suggestions and did examine the effects of leveraging the VoG score to facilitate the training of certain "atypical" examples on the fly during the training process. As a toy experiment, we trained a single layer (32 neurons) feed-forward network on the handwritten MNIST digit classification task. For comparison, we trained the model with both Stochastic Gradient Descent (SGD) and Variance Of Gradients Optimizer (VOGO) for 10 epochs using a batch size of 256. As a preliminary implementation of VOGO, we weigh each mini-batch gradient update with their respective VoG scores. Batches having a higher VoG score are relatively harder for the model to run and hence we up weight their updates in the direction of the gradient descent. The initial weights of the architectures were set to the same seed for both SGD and VoG optimizers. We observe that the VOGO achieves a lower training loss as compared to SGD. The training difference between the optimizers is reflected in the model's testing performance. Across the 5 different runs, models trained using VOGO achieve a testing accuracy of $91.77\pm0.10$% as compared to SGD which achieves almost $\approx20$% lower performance at $68.50\pm3.09$%. Notably, the performance deviation across different runs is also smaller for VOGO as compared to its counterpart.

---

### Decision · Program_Chairs · 2021-01-07
**Final Decision**

**Decision:**

Reject

**Comment:**

This paper provides a new uncertainty measure of examples called "Variance of Gradients" (VoGs); it demonstrates that VoGs are correlated with mistakes, and can be useful for guiding optimization.

On the positive side, the reviewers generally think that the ideas of this paper is nice and contribute to the research thrust in gradient-based uncertainty. In addition, the paper provides valuable empirical insights.

However, the reviewers also pointed out a few important limitations:
- A more thorough comparison to prior methods is needed to convince the readers for actual usage. There are many other methods (e.g. predicted entropy) for example difficulty estimation / classifier trustworthiness that need to be compared to.
- The stability of individual VoG scores needs to be investigated further

The authors are encouraged to address these limitations in the next iteration.